# ABLATE AND RESCUE: A CAUSAL ANALYSIS OF RESIDUAL STREAM HYPER-CONNECTIONS

**William Peng**[1*]   **Josheev Rai**[2*]   **Kevin Tseng**[3*]   **Siwei Wang**[4*]   **Sean Wu**[5]
[1]Stanford University, [2]Georgia Institute of Technology, [3]University of California, Berkeley,
[4]Independent, [5]University of Oxford

## ABSTRACT

Multi-stream transformer architectures have recently been proposed as a promising direction for managing representation collapse and the vanishing gradient problem for residual connections, yet their internal mechanisms remain unexplored. In particular, the recently introduced Manifold-Constrained Hyper-Connections (mHC) architecture posits multiple residual streams with constrained interaction, but lacks in-depth mechanistic analysis. We present the first open-source mHC language model (https://huggingface.co/wgpeng/mhc-780m) and analyze the multiple-stream architecture with a suite of representation-level metrics and causal interventions to probe how parallel streams encode and utilize information. Specifically, we introduce a systematic stream ablation-and-rescue framework that enables direct causal comparison of residual streams during inference. Through targeted pairwise interventions and controlled recovery experiments, we distinguish functional redundancy from asymmetric utilization and reveal how information is distributed across streams beyond what is observable from representational similarity alone.

## 1 INTRODUCTION

Hyper-Connections extend the standard transformer residual architecture by allowing multiple residual streams per layer, dynamically mixed through learned routing matrices (He et al., 2015; Zhu et al., 2025). Manifold-Constrained Hyper-Connections (mHC) further refines this framework by imposing geometric constraints on inter-stream mixing (Xie et al., 2026).

Despite these advances, it is unclear whether different streams encode distinct information, redundantly represent similar features, or interact asymmetrically during inference. This gap is worsened by the absence of publicly available pretrained mHC models and by the fact that most interpretability methods are designed for single-stream architectures that do not naturally extend to dynamically routed, multi-stream settings. Importantly, observational analyses alone are insufficient in this context: high representational similarity between streams does not directly imply functional interchangeability (Zhang, 2024; Hanna et al., 2023; Geiger et al., 2021; Feder et al., 2022). Understanding how information is actually used by hyper-connected models requires explicit causal interventions during the forward pass.

We borrow from black-box techniques in biological functional genomics, where ablation-and-rescue experiments are used to establish causal necessity and sufficiency. In such settings, a gene or pathway is first perturbed (e.g. via RNA interference or knockout (Echeverri et al., 2006)), producing a measurable loss of function, and the phenotype is then rescued by reintroducing the same or a compensatory functional element. Successful rescue provides strong evidence that the perturbed component plays a causal role in the observed behavior, rather than being merely correlated with it.

Rather than inferring stream importance from similarity or attribution scores alone, we introduce ablation and controlled rescue experiments to investigate stream function. By doing this, we reveal distinct regimes of redundancy, asymmetry, and complementarity between streams. Additionally, we release the first open-source trained mHC language model.

---

*Equal contribution.

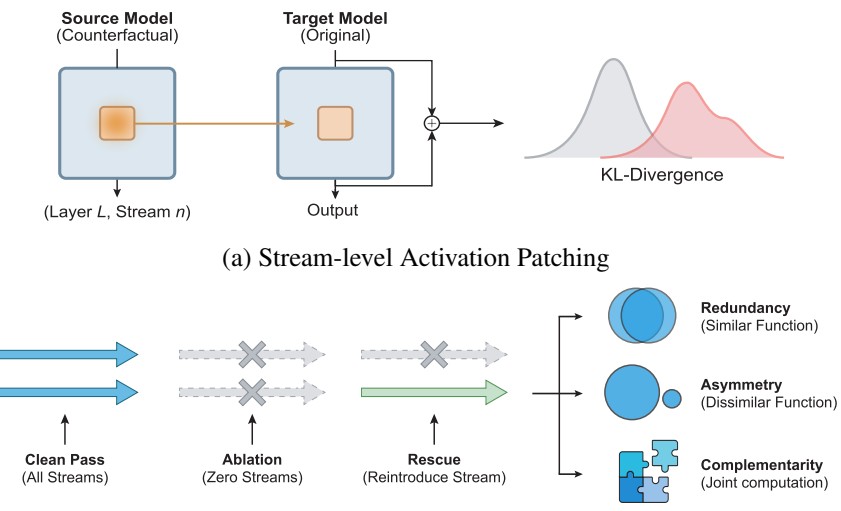

(a) Stream-level Activation Patching

(b) Ablation-and-Rescue Framework

Figure 1: **Ablation-and-rescue for causal stream analysis.** (a) Counterfactual activation patching setup. (b) Ablation-and-rescue for multi-stream architectures.

## 2 BACKGROUND

**Hyper-connections and manifold constraint.** The addition of multiple hyper-connected streams in place of a single residual connection has been shown to improve training stability and benchmark performance. In a standard transformer model, residual connections take the form $\mathbf{x}_{l+1} = \mathbf{x}_l + \mathcal{F}_l(\mathbf{x}_l)$, where $l$ is the layer index, $\mathcal{F}_l$ is the layer function, and $\mathbf{x}_l \in \mathbb{R}^d$ is the hidden state at layer $l$.

Manifold-Constrained Hyper-Connections generalize this formulation by expanding the hidden state into $n$ parallel residual streams, represented as a matrix $\mathbf{x}_l \in \mathbb{R}^{n \times d}$. Residual propagation and inter-stream mixing are governed by learned routing matrices, yielding the update

$$\mathbf{x}_{l+1} = \mathbf{H}_{\text{res}}\mathbf{x}_l + \mathbf{H}_{\text{post}}^\top \mathcal{F}_l(\mathbf{H}_{\text{pre}}\mathbf{x}_l). \tag{1}$$

Here, $\mathbf{H}_{\text{res}} \in [0,1]^{n \times n}$ is a doubly stochastic routing matrix obtained via iterations of the Sinkhorn-Knopp algorithm (Dennis & Knopp, 1967). This constraint stabilizes residual mixing by controlling operator norms and preventing uncontrolled amplification across streams.

The matrices $\mathbf{H}_{\text{pre}} \in \mathbb{R}^{1 \times n}$ and $\mathbf{H}_{\text{post}} \in \mathbb{R}^{1 \times n}$ respectively implement stream-wise aggregation and redistribution. $\mathbf{H}_{\text{pre}}$ collapses the $n$ streams into a single vector for transformation by $\mathcal{F}_l$, while $\mathbf{H}_{\text{post}}$ expands the transformed output back across streams.

**Interpretability.** Most existing interpretability techniques implicitly assume a single residual stream (Elhage et al., 2021) and therefore do not directly transfer to multi-stream architectures. We focus on representation analysis tools and causal interventions that can be adapted to multiple streams, such as CKA (Davari et al., 2022), Activation Patching (Zhang & Nanda, 2024), and targeted ablations (Li & Janson, 2024).

## 3 METHODS

### 3.1 MODEL TRAINING

Alongside supervised objectives, language models acquire a broad range of natural language capabilities (Radford et al., 2019; Gokaslan et al., 2019). We adapt the transformer block structure to incorporate Manifold-constrained Hyper-Connections and train a **781 million** parameter model comparable in size to GPT-2 Large using AdamW (Loshchilov & Hutter, 2019) and Muon optimizers (Liu et al., 2025).

In addition, we pretrain on the `dolma-v1-7` corpus, a substantially broader dataset containing a mixture of web content, academic publications, code, books, math, and encyclopedic materials (Soldaini et al., 2024).

## 3.2 Centered Kernel Alignment

To explore the encoded structures between residual streams, we utilize centered kernel alignment (CKA) which provides an interpretable visualization of geometric similarities across stream representations (Figure 2). In the foundational Hyper-connections work, Zhu et al. (2025) compares streams by layer using cosine similarity, but we opted for CKA as a more robust measure. CKA yields a similarity index between two target structures invariant to invertible linear transformations and resilient to differing random initializations (Kornblith et al., 2019), as is the case with randomly initialized stream weights. For measuring intra-layer stream relationships, we sampled per-stream residuals generated from the Pile-10k dataset for CKA (Nanda, 2022), and constructed a similarity index matrix for each layer to visualize the stream comparison scalars.

## 3.3 Activation Patching

We quantify layer–stream causal contributions to next-token prediction using counterfactual activation patching. Following symmetric token replacement (STR), we construct matched *target* prompts that replace a singular noun/verb/adjective from the source prompt, enabling causal tracing for internal activations completions (Zhang & Nanda, 2024). We evaluate patching interventions by measuring the KL-divergence of the original and patched distributions. This choice is motivated by the fact that our mHC model does not frequently rank the correct factual completion for a ROME dataset example among its top-$k$ predictions as traditionally done in causal tracing experiments (Meng et al., 2023), making accuracy-based patching criteria unstable. In particular, from the 21,919 Counter-Fact examples (Makelov et al., 2024), only 65 prompts passed the knowledge check. We instead focus on measuring patch effects on the overall token distribution between the target and counterfactual (source) model which provides a clear baseline of single stream causal contributions to token prediction.

## 3.4 Stream Ablation and Rescue

**Stream Ablation.** Let $p_\theta$ be the trained model, outputting a probability distribution over tokens, and $x = (x_1, \ldots, x_T)$ be the input sequence to the model. At layer $\ell$, for token $t \in [1, T]$ and stream $s \in [0, n-1]$, the residual stream activation is $\mathbf{x}_{t,s}^{(\ell)} \in \mathbb{R}^d$. We first run an unperturbed forward pass, caching and freezing the Hyper-Connection mixing matrices, and storing $\mathbf{x}_{t,s}^{(\ell)}$. For a stream pair $(i, j)$, our ablation experiment defines each $\tilde{\mathbf{x}}_{t,s}^{(\ell)}$ as follows:

$$\tilde{\mathbf{x}}_{t,s}^{(\ell)} = \begin{cases} \mathbf{0}, & s \in \{i, j\}, \\ \mathbf{x}_{t,s}^{(\ell)}, & \text{otherwise.} \end{cases} \tag{2}$$

Ablation impact is measured by the mean token-wise KL divergence, where $(-i, -j)$ denotes ablation of streams $i$ and $j$. In our experiments, we calculate $p_\theta$ using temperature 1.

$$\mathcal{L}_{\text{KL}}^{(-i,-j)} = \mathbb{E}_{x,t}\big[\text{KL}(p_\theta(y_t \,|\, x) \,\|\, p_\theta^{(-i,-j)}(y_t \,|\, x))\big]. \tag{3}$$

**Targeted Rescue.** To test recoverability, we restore ablated stream $i$ using cached residuals while keeping the other ablated, yielding $p_\theta^{(+i,-j)}$. Rescue is reported as the fractional KL reduction relative to full ablation,

$$\text{Recovery}(+i, -j) = 1 - \frac{\mathcal{L}_{\text{KL}}^{(+i,-j)}}{\mathcal{L}_{\text{KL}}^{(-i,-j)}}. \tag{4}$$

By expanding this across all possible stream pairs, we construct a global rescue matrix that distinguishes redundant, asymmetric, and complementary stream contributions.

## 4 RESULTS

### 4.1 REPRESENTATIONAL SIMILARITY ACROSS STREAMS

The middle layers of the model form a visually distinctive checkerboard-like pattern across their CKA matrices (Figure 4), suggesting the model learns a representational divide of streams into two groupings based on similarity. These feature groups manifest in full by Layer 12 and gradually diminish as distinctness between streams collapses by the final layer.

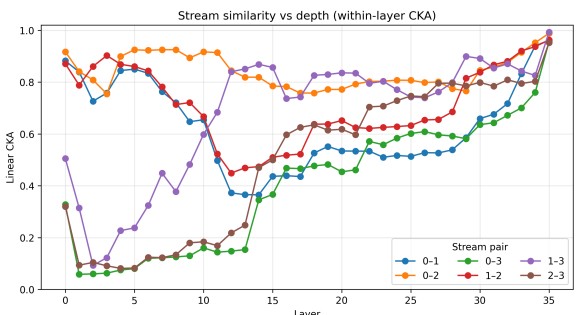

Figure 2: **Within-layer similarity.**

### 4.2 STREAM-LEVEL CAUSAL CONTRIBUTIONS VIA ACTIVATION PATCHING

Activation patching surfaced a distinct asymmetry in residual stream contributions to the final token distribution (Figure 3). Notably, streams (0, 2) demonstrate higher sensitivity to individual token context during inference than streams (1, 3). Depth-wise patching yielded low or diminishing patch effects with the exception of stream 2 which maintains strong patching sensitivity deep into the mid layers of the model.

### 4.3 FUNCTIONAL REDUNDANCY AND ASYMMETRY VIA RESCUE

Across layers with high cross-stream CKA, we observe distinct functional regimes. In one, streams exhibit mutual recoverability: streams 0 and 2 can each independently restore much of the KL divergence caused by ablation, indicating functional redundancy beyond representational similarity. In contrast, other stream pairs show clear asymmetries. For instance, rescuing stream 3 restores KL-divergence by 15.86% more than rescuing stream 1 (Table 1). This indicates an imbalance in functional contribution, despite relatively high representational similarity, highlighting that CKA alone cannot distinguish between active utilization and passive redundancy. Complementarity, where information is jointly distributed across streams, is less prevalent in this model configuration. We do not observe cases where neither stream alone is sufficient to restore performance while their combination is.

| | Recovered stream | | | |
|---|---|---|---|---|
| **Ablated stream** | **0** | **1** | **2** | **3** |
| 0 | – | 58.69 | 74.78 | 66.45 |
| 1 | 84.40 | – | 81.10 | 82.42 |
| 2 | 80.61 | 58.47 | – | 71.51 |
| 3 | 71.14 | 66.56 | 72.80 | – |

Table 1: **Mean rescue performance across residual streams.** Each entry reports the average percentage of KL-divergence recovery over layers when ablating a pair of streams and selectively rescuing only one of them . Diagonal entries are undefined since a pair of identical streams cannot be independently ablated and rescued.

## 5 CONCLUSION

Our results highlight stream asymmetries and show that high representational similarity does not imply functional interchangeability, motivating rescue-style causal experiments for analyzing redundancy and asymmetry in multi-stream architectures.

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

## A    APPENDIX

**Overview.**    The following supplementary analyses reinforce the main text's causal claims about residual stream behavior in mHC models. Together, these results support three central conclusions: (i) causal influence is sharply concentrated within specific layers and streams, (ii) representational similarity is informative but insufficient to predict functional interchangeability, and (iii) explicit interventions reveal structured regimes of redundancy and asymmetry that are otherwise obscured by observational metrics.

**Layer-wise causal localization.**    We begin by examining where causal control over the output distribution is concentrated using activation patching. As shown in Figure 3, causal influence is not uniformly distributed. Notably, stream 2 maintains strong influence deep into the network, contrasting with the relative passivity of stream 1. This stratification motivates the use of pairwise causal interventions to uncover the structure underlying these contributions.

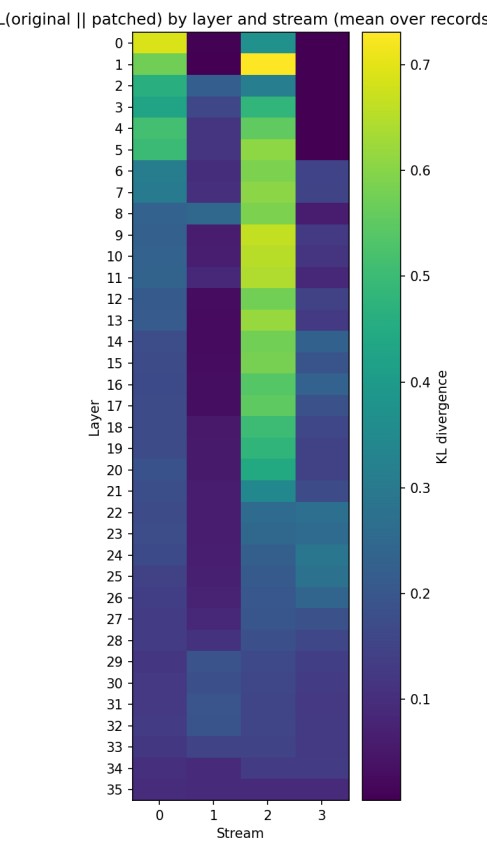

Figure 3: **Layer–stream causal sensitivity via activation patching.** Mean KL divergence between baseline and patched logits when one (layer, stream) activation is injected from source to target run. Lighter values indicate stronger causal effect.

**Emergent stream structure via CKA.**    To assess how representations evolve and align across residual streams, we compute intra-layer CKA matrices (Figure 4) and inter-layer CKA heatmap (Figure 5). In the middle layers, streams consistently bifurcate into two highly similar subgroups. This structure dissolves in later layers as representations converge. Inter-layer CKA reveals two distinct regions of high similarity, suggesting stable representational phases between the early and mid-to-late stages of the model.

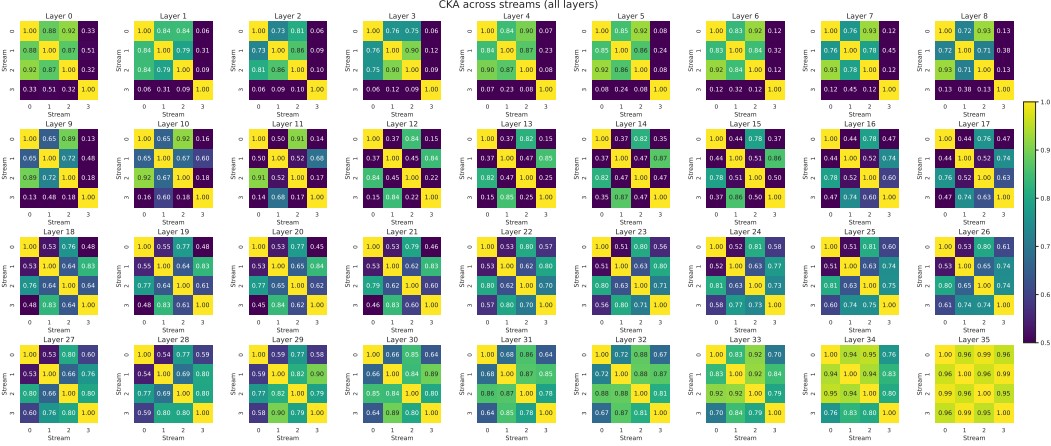

Figure 4: **Within-layer CKA similarity matrices across depth.** Middle layers show clear block structure, reflecting soft partitioning into redundant stream subgroups.

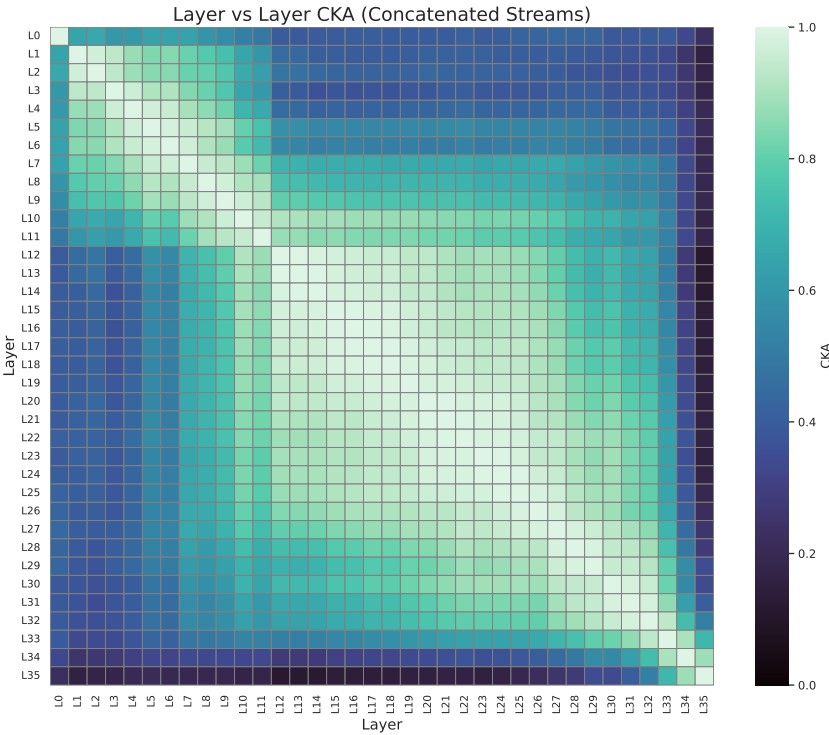

Figure 5: **Inter-layer CKA with streamwise concatenation.** Layers evolve gradually in their representational geometry.

**Routing dynamics across depth.** We examine how the learned routing matrices evolve with depth. As shown in Figure 6, both the Frobenius norm and variance of $\mathbf{H}_{\text{post}}$ increase with layer index, suggesting that downstream layers amplify and diversify the outputs of intermediate stream aggregation. In contrast, $\mathbf{H}_{\text{pre}}$ and $\mathbf{H}_{\text{res}}$ remain stable, indicating that only the post-aggregation redistribution becomes more diffuse as representations are pushed toward the output.

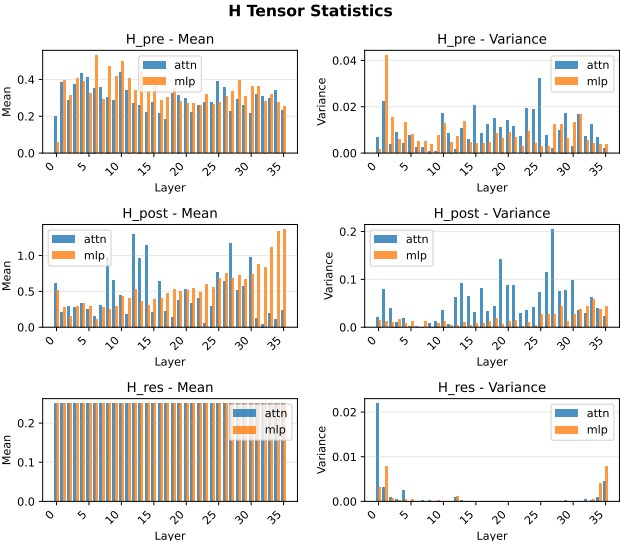

Figure 6: **Routing dynamics across depth.** Upward trend in $H_{post}$ reflects growing inter-stream dependence, aligning with observed causal convergence.

**Redundancy and asymmetry in rescue.** Rescue experiments isolate the degree to which one stream compensates for another. Figure 7 shows that stream pair (0,2) exhibits high mutual rescue, suggesting redundancy. Others, such as (1,3), show asymmetric recovery where stream 3 reliably compensates for stream 1, but not vice versa. These patterns indicate that residual streams may play different roles during the forward pass, despite comparable representations.

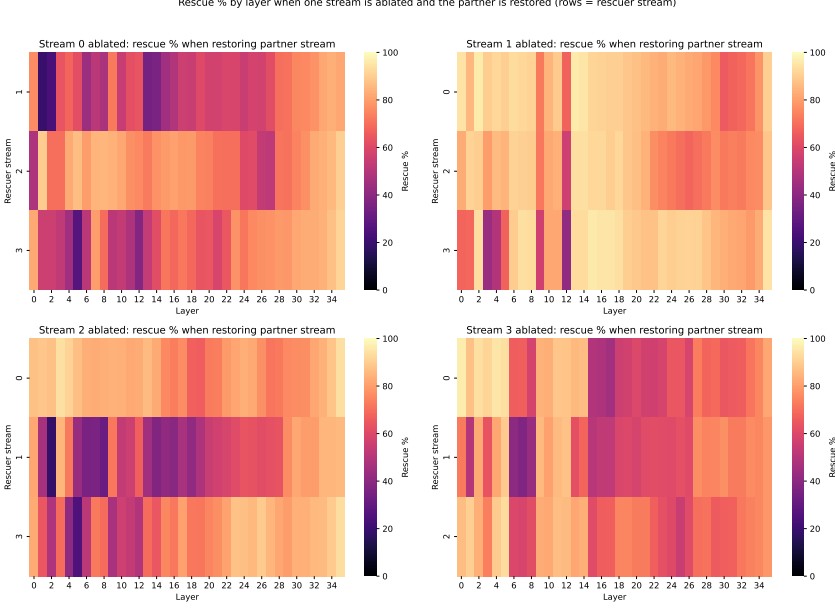

Figure 7: **Layer-wise rescue performance by stream.** Rescue values are defined as percentage KL reduction from full ablation. High scores indicate functional redundancy; low scores suggest complementarity or general asymmetry.

**Full pairwise comparisons.** To visualize recovery regimes across all stream pairs, Figure 8 reports the distribution of KL scores from joint ablation and single-stream rescue. Symmetric recovery suggests redundant encoding, while skewed or weak rescue indicates directional or complementary encoding. Stream pair (0,2) shows tight symmetric rescue, while pair (1,3) shows stream 3 dominating recovery.

Rescue experiment box plots: KL when restoring one ablated stream (all pairs)

Figure 8: **Distribution of rescue effects across stream pairs.** Boxplots summarize KL recovery values across layers, revealing asymmetric and symmetric recovery patterns.

**Quantifying asymmetric utility.** To directly contrast symmetric and asymmetric stream pairs, Figure 9 plots the per-layer rescue difference between (0,2) and (1,3). The near-zero values for (0,2) suggest interchangeable function, while consistent positive differences for (1,3) indicate persistent asymmetry. This validates our central claim: high representational similarity does not imply causal interchangeability.

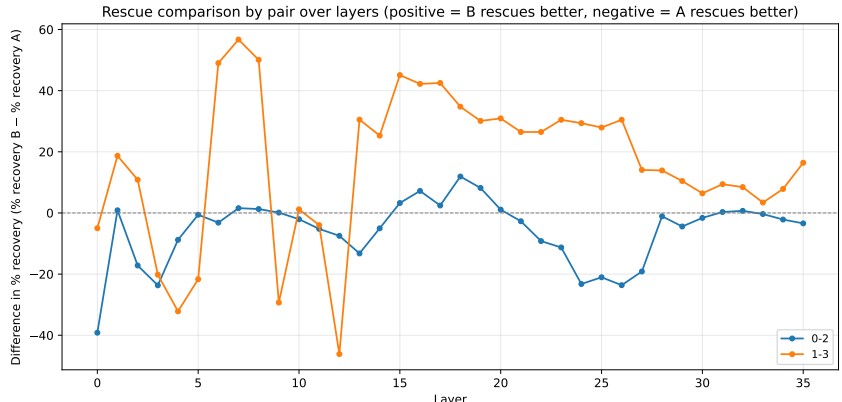

Figure 9: **Layer-wise rescue asymmetry.** Positive values indicate that the second stream in a pair is more effective at recovering the joint ablation. Stream 3 consistently dominates stream 1 despite high CKA.

| Hyperparameter | Value |
|---|---|
| *Architecture* | |
| Architecture | GPT-2 (decoder-only Transformer) |
| Parameter count | 781M |
| Layers | 36 |
| Hidden dimension | 1280 |
| Attention heads | 20 |
| Head dimension | 64 |
| Embedding dimension | 1280 |
| Context length | 1024 |
| Vocabulary size | 50304 |
| Residual streams | 4 |
| Hyper-connection type | mHC (Manifold-Constrained) |
| Dropout | 0.0 |
| Bias | True |
| *Training* | |
| Optimizer | AdamW + Muon |
| Learning rate | $3 \times 10^{-4}$ |
| Min learning rate | $3 \times 10^{-5}$ |
| Schedule | Cosine decay |
| Weight decay | 0.1 |
| $\beta_1$, $\beta_2$ | 0.9, 0.95 |
| Gradient clipping | 1.0 |
| Warmup steps | 200 |
| Batch size | 0.5M tokens |
| Training steps | 10,000+ |
| *Data* | |
| Dataset | Dolma-v1_7 |
| Tokens seen | $\sim$3.18B |

Table 2: **Model and training hyperparameters.** Configuration of our 781M parameter mHC-GPT2 model. Architecture augments GPT-2 with 4 Manifold-Constrained residual streams.

