# OpenReview forum: "Ablate and Rescue: A Causal Analysis of Residual Stream Hyper-Connections"
_ICLR.cc/2026/Workshop/Sci4DL — Sci4DL 2026_

### Official Review · Reviewer_Sg93 · 2026-02-15

**Fit:** 2
**Significance:** 2
**Confidence:** 2

**Summary:**

This paper presents a causal analysis of Manifold-Constrained Hyper-Connections (mHC), a novel multi-stream transformer architecture whose internal mechanisms remain underexplored. The authors introduce an ablation-and-rescue framework inspired by functional genomics, alongside representation-level analyses using CKA and activation patching. Their experiments show that residual streams with high representational similarity can exhibit asymmetric or redundant functional roles, demonstrating that similarity alone does not imply interchangeability. Overall, the work provides a principled causal approach to analyzing information flow in multi-stream transformer models.

**Strengths:**

- **Relevant and generalizable methodology**: The proposed ablation-and-rescue framework offers a clear and generalizable methodology for causal analysis in multi-stream architectures, extending existing interpretability tools largely designed for single-stream transformers.
- **Principled study and evaluation of different techniques**: The study is methodologically principled and evaluates multiple complementary techniques (including CKA, activation patching, and targeted ablation) to probe both representational and functional properties of residual streams.
- **Inspiration from other scientific domains**: The methodology draws inspiration from functional genomics, particularly ablation-and-rescue experiments, a well-motivated interdisciplinary analogy that strengthens their proposed method.

**Suggestions:**

- **Better contextualization of mHC models**: The authors motivate the work in part by the “absence of publicly available pretrained mHC models.” Given that the original mHC paper was published in January 2026, this absence seems expected rather than a significant limitation. Similarly, the contribution of releasing the “first open-source trained mHC language model,” while useful, may be overstated. Rephrasing these points to better reflect the recency of the architecture would strengthen the positioning of the contribution.
- **Computational resources**: The paper would benefit from reporting the computational resources required to train the 781M-parameter mHC-GPT2 model (e.g., number and type of GPUs, training time). This information is important for reproducibility and for assessing the practical feasibility of adopting the proposed approach.
- **Further analysis of results**: If space permits, the results section could include deeper analysis of the findings and their meaning. As currently presented, it is somewhat difficult to extract a concise takeaway regarding the broader implications or practical applicability of the observed redundancy and asymmetry patterns.
- **Limitations and future work**: it would be important to acknowledge the limitations of the current work and relevant research avenues for the future.
- (Minor) Line 106 is missing the AdamW citation

---

### Official Review · Reviewer_dbCN · 2026-02-26

**Fit:** 3
**Significance:** 2
**Confidence:** 2

**Summary:**

The paper investigates the internal mechanisms of mHC architectures which extend transformers with multiple residual streams. The authors train the first open source mHC language model with 781M parameters (GPT-2 scale) and apply 3 analysis methods:
- CKA for representational similarity
- activation patching for causal contribution of individual streams
- a novel "ablate and rescue" framework inspired by biological functional genomics

The rescue framework ablates pairs of streams then selectively restores one which enables distinctions between functional redundancy, asymmetry and complementarity. Their key findings include that high representational similarity (CKA) does not imply functional interchangeability, streams exhibit structured asymmetries where some can compensate for others but not the other way around, and streams bifurcate into two similarity subgroups in the middle layers.

**Strengths:**

- The methodology is novel and the ablation-and-rescue framework is a creative adaptation from genomics. It provides causal evidence that goes beyond observational similarity metrics like CKA, which is a meaningful contribution
- The paper is well structured with clean formalisation of the ablation and recovery metrics. The distinction between redundancy, asymmetry and complementarity is well motivated and clearly implemented
- The demonstration that high representational similarity does not imply functional interchangeability is a valuable insight that challenges a common assumption in interpretability work
- Training and releasing an open source mHC model is a practical contribution that enables future work on multi stream architectures

**Suggestions:**

- The paper does not argue why mHC architectures matter or why the reader should care about understanding their internal mechanisms. The introduction states they are "promising" but provides no evidence of concrete advantages over standard transformers. Without this motivation the reader has little reason to invest in the technical details that follow
- The paper does not clearly explain what the rescue framework provides over simpler alternatives. Ablating both streams i and j and then restoring i (with cached activations) provides a similar state to just ablating j. The key difference is that rescue uses i's clean activations (from the unperturbed model) rather than letting i naturally adapt. This distinction is not discussed. Additionally, the same information can be approximated by comparing single ablation and double ablation KL values, which avoids the cached activation confound. The paper should jutify why cached clean activations are the right counterfactual to use
- All findings come from a single model with 4 streams which only passes 65 of ~21k CounterFact knowledge checks. It is unclear whether the observations generalise to other mHC configurations, scales or other models. No comparison to a standard single stream transformer is provided which makes it hard to assess what the multi stream structure actually gives us
- The observed functional asymmetry could reflect different routing weights rather than meaningful differences in what the streams compute. The paper does not disentangle these explanations.
- Generally throughout the paper the presentation could be clearer, there is too much focus on technical detail at the expense of helping the reader understand the implications. For example, CKA is introduced without justification for why it's the right metric, results are stated without interpretation (the checkerboard pattern is reported without an explanation for what was expected and why it matters), the first result references an appendix figure (Figure 4) while Figure 2 is in the main text and isn't references. Also minor issue with missing citation for AdamW

---

### Official Review · Reviewer_6KL4 · 2026-02-27

**Fit:** 2
**Significance:** 2
**Confidence:** 1

**Summary:**

This paper studies Manifold-Constrained HyperConnections (mHC) — a multi-stream residual architecture for Transformers—and proposes an "ablate-and-rescue" causal intervention framework to distinguish representational similarity from functional interchangeability across residual streams. The authors also claim to release the first open-source pretrained mHC language model and present analyses using CKA, activation patching, and pairwise stream ablation + targeted rescue measured via KL divergence.

**Strengths:**

The paper’s framing is a clear strength: it explicitly recognizes that observed representational similarity between streams is not sufficient to conclude functional interchangeability, and therefore motivates causal interventions during the forward pass rather than relying purely on correlational or geometric diagnostics. By grounding the analysis in what changes in the model's behavior when a stream is perturbed-and contrasting this with what similarity metrics suggest—the paper sets up an appropriate methodology for answering mechanistic questions about multi-stream architectures that observational analyses alone cannot resolve.

**Suggestions:**

The paper should broaden its evaluation beyond the single 4-stream mHC configuration and single 781M-parameter checkpoint trained for 10k+ steps / ~3.18B tokens. I recommend replicating the core findings across multiple random seeds, different numbers of streams, and at least one additional model scale to test whether the observed patterns are stable architectural phenomena rather than specific to one training run.

---

### Meta-Review · Area_Chair_R4QW · 2026-03-01

**Recommendation:** Accept

**Metareview:**

The paper is well written, easy to follow and makes significant contributions. I recommend acceptance.

---

### Decision · Program_Chairs · 2026-03-02

Accept